# Believe It or ‘*Nut*’: Why It Is Time to Set the Record Straight on Nut Protein Quality: Pistachio (*Pistacia vera* L.) Focus

**DOI:** 10.3390/nu15092158

**Published:** 2023-04-30

**Authors:** Emma Derbyshire, Jennette Higgs, Mary Jo Feeney, Arianna Carughi

**Affiliations:** 1Nutritional Insight, Epsom KT17 2AA, UK; 2Food To Fit, Kingston-upon-Thames, London KT1 4AE, UK; jennette@foodtofit.com; 3California Agricultural Boards, Los Altos Hills, CA 94024, USA; 4American Pistachio Growers, Fresno, CA 93720, USA

**Keywords:** pistachio, protein demands, protein diversity, protein quality, sustainability

## Abstract

There are growing public health movements to transition towards diets that are plant-based. However, confusion exists with concerns that plant-based proteins (including nuts) may be inferior with respect to protein quality. The present publication evaluates the evolution of protein quality concepts and explains the protein science related to pistachios. Pistachio nuts are a plant-based complete protein providing all nine EAAs in addition to an array of nutrients and phytochemicals. They have a PDCAAS of 73 and 81%, (raw and roasted pistachios, respectively), higher than that of many other tree nuts. From an environmental perspective transitioning towards plant-based diets (including nuts) could have potential to reduce total/green water footprints. Dietary guidelines are evolving yet nuts such as pistachios do not always have a clear place within these. Now appears to be a pertinent time to look at protein quality from the perspective of whole daily diets and dietary patterns, factoring in both health and environmental outcomes. Given updated modes of thinking, nuts such as pistachios have an important role to play in terms of providing ready-to-eat, good-quality, plant-based protein within daily diets.

## 1. Introduction

Plant-based diets are growing in popularity globally for an array of reasons which includes concerns for human and planetary health [1,2]. Concepts of what constitutes a ‘plant-based’ diet vary considerably, with present definitions ranging from the elimination of all animal products to diets including dairy, fish and meat in variable amounts [3]. There have been concerns, however, that plant-based diets provide lower levels of the nine essential amino acids (EAA), including specific amino acids, such as leucine, sulphur amino acids and lysine [4,5,6,7]. Compared with animal-derived proteins it is further alleged that plant-proteins have less of an anabolic effect, due to their lower digestibility and the amino acids being directed towards oxidation rather than muscle protein synthesis (MPS) [7]. It is well appreciated that within the general population misconceptions exist about what constitutes health eating, even more so with goalposts ever shifting in terms of what defines a healthy diet [8,9]. From a consumer behaviour perspective, enjoyment of meat, unwillingness to make dietary changes and health concerns could act as potential obstacles to the shift towards plant-based diets [10]. From a historical standpoint across pre-agricultural hunter-gatherer societies, meat was regarded as a ‘highly prized food’ and today still has strong associations with masculinity [11].

The EAT-Lancet 2019 Food in the Anthropocene report was revolutionary in that it accentuated the need to review food systems and their dual impacts on health and planetary outcomes and included a modelled healthy reference diet that was predominantly plant-based [12]. The EAT-Lancet healthy reference diet advised that this “largely consists of vegetables, fruits, whole grains, legumes, nuts, and unsaturated oils” and as a key message concluded that the transformation to healthy diets by 2050 will require substantial dietary shifts, including a greater than 100% increase in consumption of foods, such as nuts, fruits, vegetables and legumes [12]. Within this guidance, peanuts were listed as a separate protein food source within the legumes category (with a possible daily intake range of 0–75 g) and tree nuts, which would encompass pistachios, were also listed as a separate category (possible intake of 25 g/day) [12]. The EAT Lancet dietary guidance therefore recognises the role of nuts as a key plant protein source.

Shifting to a more plant-based diet changes the dietary profile, reducing intakes of predominant (animal) protein sources, which has led to some questions regarding the adequacy of protein quality [13]. One view is that we may be thinking too ‘narrowly’ about dietary protein forms and that it is the overall amino acid composition (diversity) of the diet that determines protein adequacy [14,15]. Present definitions of protein quality are becoming obsolete and misleading, focusing solely on essential amino acid profiles for single foods and falling short when it comes to considering extended influences on human and environmental outcomes [14]. These dated definitions are therefore becoming misaligned and out of sync with movements towards the concept of protein diversification, which is highly relevant to modern-day nutrition and environmental policies [16,17,18,19].

The current publication focuses in on pistachios and explains how this example nut can contribute to daily protein intakes and amino acid profiles, as part of the shift to greater plant-based protein sources. Pistachio nuts (*Pistacia vera* L.) are tree nuts with epidemiological and clinical studies increasingly reporting positive health outcomes attributed to tree nut consumption [20]. Pistachio nuts provide protein, fibre, monounsaturated fatty acids, an array of vitamins, minerals, phytochemicals, and polyphenols [20,21,22]. Pistachios align with European Union/United Kingdom nutrition claims qualifying them as being ‘high in’ fibre, monounsaturates, copper, chromium, vitamin B6, thiamine, manganese, phosphorus and potassium and ‘a source’ of protein, vitamin E, K, folic acid, riboflavin, magnesium, iron, zinc and selenium [23,24,25]. Yuan et al. (2022) analysed the phytochemical profile of raw and roasted American pistachios finding that gentisic acid and catechin were predominant phenolics, with raw and roasted pistachios possessing similar total antioxidant activity levels and cellular antioxidant activity [26]. The nutritional and phytochemical profile of pistachios is summarised in Table 1. Relevantly, present-day pistachio consumption has been linked to a spectrum of health benefits including favourable antioxidant and ant-inflammatory actions, endothelial function and glycaemic control [27]. Meta-analytical evidence linking pistachio consumption to improved cardiometabolic risk factors (fasting blood sugar, insulin, systolic blood pressure and lipid profiles) without increasing body mass index also looks favourable [28].

From an environmental perspective it is foreseen that the future ‘protein-economy’ will be mixed [17,18]. As the global population surges from 7.2 billion to a projected 10 billion people by 2050, plant-sourced protein will have an important role to play in meeting protein demands [2,29]. These data starkly emphasise the need for diversification of food proteins. Given this, the present paper emphasises the need to review the concept of protein quality and rethink the role of plant protein, focusing on pistachios as an example to help meet protein needs within the context of health promoting dietary patterns.

**Table 1 nutrients-15-02158-t001:** Nutrients in 100 g of Tree Nuts (Pistachios).

Nutrients and Phytochemicals (Units)	Nutrient Content per 100 g (Dry Roasted Pistachios, without Salt Added)
**Nutrients**	
Energy (kJ/kcal)	2390/572
Protein (g) *	21
Total Fat (g)	46
Saturated Fat (g)	5.6
Monounsaturated Fat (g)	25
Polyunsaturated fat (g)	13
Cholesterol (mg)	0
Carbohydrate (g)	28
Fibre (g)	10
Calcium (mg)	107
Iron (mg)	4
Magnesium (mg)	109
Phosphorus (mg)	469
Potassium (mg)	1010
Sodium (mg)	6
Zinc (mg)	2.3
Copper (mg)	1.3
Manganese (mg)	1.2
Selenium (mcg)	10
Chromium (mcg)	39 ^a^
Vitamin C (mg)	3
Thiamine (mg)	0.7
Riboflavin (mg)	0.2
Niacin (mg)	1.4
Pantothenic acid (mg)	0.5
Vitamin B6 (mg)	1.1
Folate (mcg)	51
Choline, total (mg)	71
Betaine (mg)	0.8
Vitamin B12 (mcg)	0
Vitamin A (IU)	266
Vitamin K (mcg)	13.2
**Vitamin E**	
Tocopherol, alpha (mg)	2.2
Tocopherol, beta (mg)	0.1
Tocopherol, gamma (mg)	23.4
Tocopherol, delta (mg)	0.6
**Phytochemicals Carotenoids and Phytosterols**	
Carotene, beta (mcg)	159
Carotene, alpha (mg)	0
Cryptoxanthin, beta (mcg)	0
Lutein + zeaxanthin (mcg)	1160
Stigmasterol (mg)	2
Campesterol (mg)	10
Beta-sitosterol (mg)	210
**Total phenolics 448 mg GAE/100 g,**	
Gallic acid (mg)	18.1 ^b^
Protocatechuic acid (mg)	3.25 ^b^
Gentisic acid (mg)	212 ^b^
**Flavonoids Total 144 mg GAE/100 g**	
Catechin (mg)	33.8 ^b^
Epigallocatechin (mg)	6.26 ^b^
**Other Polyphenols**	
Catechol (mg)	7.88 ^b^

Key: Source of claim possible (light shading); High in claim possible in the European Union (dark shading). * May only be made where at least 12% of the energy value of the food is provided by protein [30]. Data extracted from the U.S. Department of Agriculture. Agricultural Research Service. FoodData Central. 2019 NDB Number: 12152 [31]. ^a^ Cabrera et al. (2003) [32]; ^b^ Yuan et al. (2022), data extracted for total values (bound + free phenolics) for roasted pistachios [26]. g = gram; GAE = Gallic Acid Equivalents; mcg = microgram; mg = milligram; IU = International Units; RI, Reference Intake.

## 2. The Evolution of Protein Quality Concepts

Dietary protein comprises amino acids joined by peptide bonds, which are hydrolysed in the lumen of the digestive tract by peptidases and proteases to yield amino acids, dipeptides and tripeptides [33]. Amino acids can be classified into two distinct categories—essential (indispensable) amino acids (EAAs) and non-essential (dispensable) amino acids (NEAAs) [34,35]. It is ‘essential’ that EAAs are obtained from exogenous dietary sources as unlike NEAAs they cannot be synthesised endogenously—the human body lacks metabolic pathways to produce these [34,36]. There are nine EAAs: leucine, isoleucine, valine, phenylalanine, threonine, tryptophan, methionine, lysine and histidine [36]. Within these, leucine, isoleucine and valine have aliphatic side chains, thus are known as branched-chain amino acids (BCAAs) [34]. The BCAAs have received increasing attention due to their ability to induce protein synthesis [34]. EAAs are also available from incomplete proteins (those not providing all nine EAAs in adequate amounts), and in these instances the term ‘limiting amino acid’ is used to describe the EAA present in the lowest quantity relative to a reference food protein or requirement pattern [36]. This term can also be used when a specific EAA does not meet minimal requirements for humans [36,37].

To determine the quality of a protein source, one approach is to calculate a Protein Digestibility-Corrected Amino Acid Score (PDCAAS) [38]. The PDCAAS measures the quality of a protein and is based on amino acid requirements of humans, adjusted for digestibility, and can be compared against a reference population [38]. The Food and Agriculture Organisation (FAO) of the United Nations and World Health Organisation (WHO) adopted the process of PDCAAS scoring in 1990 and regarded this as the ‘gold standard’ and preferred method to determine protein quality [39,40]. The PDCAAS is typically expressed as a percentage (mg of limiting amino acid in 1 g of test protein/mg of same amino acid in 1 g of reference protein) x true digestibility percentage), thus true protein digestibility is the difference between nitrogen ingested and faecal nitrogen excreted [39,41,42].

Although the PDCAAS is a useful and rapid method, scientists have questioned its accuracy as discrepancies have been observed when comparing derived values with other methods such as the Protein Efficiency Ratio (PER; a measure of body weight gained divided by the weight of protein ingested) [41]. Equally, when adopting PDCAAS methods animal-derived proteins (>95% digestibility) appear to rank more favourably than plant proteins (50–80% digestibility) due to a higher distribution of the nine essential amino acids and digestibility [43,44].

Given these limitations, an updated scoring system known as the Digestible Indispensable Amino Acid Score (DIAAS) was advocated in 2013 by the FAO and recognised as an alternative way to determine protein quality [45,46,47]. The DIAAS addresses some of the inadequacies of the PDCAAS method by considering the ileal digestibility coefficients of individual amino acids, being based on a growing pig model rather than a rodent model and not truncating values at an upper limit of 100 [14,48]. However, the DIASS, just like the PDCAAS, continues to disregard broader issues of relevance, such as longer-term measures of health and environmental implications [16]. The method is also based on animal models and thus not directly applicable to humans [49]. Furthermore, it fails to: (1) translate differences in nitrogen-to-protein conversion factors between plant- and animal-based foods, (2) has limited representation of plant-based foods within the scoring framework, (3) fails to recognise the increased digestibility of heat-treated/processed plant-based foods and (4) focuses on individual isolated foods rather than the food matrix [49]. Subsequently, it is argued that the DIAAS is being used out of context and should be avoided when investigating plant-based protein quality [49].

Alongside these limitations, the term ‘protein quality’ can be used rather broadly in public, scientific and indeed legislative domains. For example, clear distinctions are needed between the term ‘protein quality’ (the amount of amino acids needed to maintain health and its digestibility) versus foods that are ‘a source’ of protein, which relates specifically to the absolute protein content in a specified food [25]. Different countries’ systems tend to place less emphasis on protein quality for labelling and claims. As shown in Table 2, the European Union and United Kingdom determine protein by percentage energy and others, such as Australia and New Zealand, refer to weight of protein per serving. In the United States and Canada, protein content claims on labels must be substantiated based on both protein quantity and quality. Canada uses the PER method and most recently also the PDCAAS to measure the protein quality of foods [42]. The U.S. Food and Drug Administration (FDA) requires the PDCAAS method, as described in the 1991 FAO report, to show a percent daily value for protein on the nutrition facts panel as well as the substantiation of protein content claims on labels [50].

## 3. Protein in Pistachios

The PDCAAS of only a few nuts has been evaluated. In 2020 a standardised ileal digestibility study measured both the DIAAS and PDCAAS score of raw and roasted pistachio nuts [56]. Raw and roasted pistachio nuts had a PDCAAS of 73% and 81%, respectively, calculated for children 2–5 years, with the limiting amino acid being threonine in the calculation. The DIAAS was 86 and 83 for raw and roasted pistachio nuts, respectively, calculated for children older than 3 years, adolescents and adults, with the limiting amino acid being lysine [56]. Taken together, these results demonstrated that both raw and roasted pistachio nuts had a DIAAS greater than 75, thus were recognised as being ‘good quality’ protein sources [56].

The PDCAAS of other nuts has also been determined. The protein digestibility of cashew nuts, peanuts and Brazil nuts and their PDCAAS have been calculated to be 90.3%, 69.3% and 63.3%, respectively [57]. For raw almonds, a PDCAAS between 44.3 and 47.8% was equated for children aged 2–5 years [58]. For walnuts, a PDCAAS of 46% has been calculated for children aged 3–10 years [38]. In most of these studies the specified limiting EAA was lysine [56,57,58]. Researchers have recently compiled a database of amino acids in plant-sourced foods [59]. The amino acid found to be present in the greatest concentration for both raw and roasted pistachio nuts has been found to be arginine, with leucine being the second most abundant in concentration [56]. Other research shows that free amino acids represented ≤3.1% of total AAs in pistachio nuts, soybeans, corn grains, white rice and wheat flour and 34.4% and 28.5% in potatoes and sweet potatoes, respectively [59]. Subsequently, as shown in Table 3, a PDCAAS of 73 and 81%, respectively, for raw and roasted pistachios is higher than scores calculated for almonds, Brazil nuts, peanuts, pecans and walnuts. It is also higher than PDCAAS derived for white rice, chickpeas, red kidney beans and certain lentils (whole green, red and yellow split lentils). The PDCAAS of roasted pistachios was only marginally lower than that of chicken, beef and egg powder. This demonstrates that plant-based pistachios are a balanced and good quality protein source.

There is increased recognition, however, that we should be focusing on the ‘totality of diets’ rather than whether or not individual foods provide all nine EAA [14]. People ingest mixed diets, thus it is the amino acid composition of the overarching diet that will determine protein adequacy [14]. Thus, the concept of protein diversification, with protein from a range of food sources beyond those that are animal-derived, is gaining interest [16,17].

As demonstrated in Table 4 different foods provide an assortment of different amino acids. Pistachio nuts are a complete protein providing all nine EAAs along with 4.3 g glutamic acid per 100 g (comparable with beef brisket) and 2.1 g arginine per 100 g (comparable with chicken). After braised beef brisket and cooked chicken, pistachios provide the next highest levels of total EAAs (7.9 g/100 g). Pistachios also provide a higher level of branched chain amino acids compared with other nuts and foods such as brown rice and lentils. Figure 1 further demonstrates that pistachios are an important provider of EAAs.

Mariotti and Gardner (2019) recently explained there have been concerns that amino acid intakes from vegetarian diets are often perceived as being inadequate, but the integration of nuts, seeds, and legumes within diets can be sufficient in terms of achieving full protein adequacy in adults, provided that energy needs are being met and a variety of foods are being consumed [66]. Pistachios are therefore positioned as a useful protein source, delivering good-quality plant-based protein, fibre, a range of healthy fats, including mono- and polyunsaturated fatty acids whilst being low in saturated fatty acids and providing an array of micronutrients and bioactive compounds [25,26,67].

## 4. Narrowing the Gap

### 4.1. Habitual Intakes of Tree Nuts

As shown in Table 5, several surveys and studies have measured habitual dietary intakes of tree nuts, with some reporting specifically on pistachios. O’Neil et al. (2015; 2010) analysed U.S. National Health and Nutrition Examination Survey (NHANE) data, finding in both analyses that tree nut consumption was low with only 5.5–8% of the population consuming these [68,69]. Amongst those consuming nuts, usual intakes were 44.3 ± 1.6 g/day [69] thus only just aligned with Food and Drug Administration evidence that eating 42.5 g per day may reduce heart disease risk [70].

Data from the European Prospective Investigation into Cancer and Nutrition (EPIC) study [71] demonstrated that only 0.4% of men and 0.7% of women consumed pistachios with reported portion size intakes of 22 g/d and 23.1 g/d.

**Table 5 nutrients-15-02158-t005:** Intakes of Tree Nuts–Key Studies.

Country	Name of Study/Survey	Sample Size and Age	Level of Intake	Key Findings
Van den Brandt and Nieuwenhuis (2018) [72] Netherlands	Netherlands Cohort Study	n = 62,573 women, 55–69 years	Tree nut intakes were 1.0 (SD 3.9) g/day.	Total nut intake was significantly inversely related to oestrogen receptor breast cancer risk, with HR 0.55 (95% CI 0.33–0.93) for those consuming at least 10 g nuts/day versus non-consumers.
Tan et al. (2021) [73] United States	NHANES (2011–12 and 2013–14) cohorts	n = 1848 (≥60 years)	Moderate nut intake (15.1–30.0 g/d) may be sufficient for better cognitive performance.	Moderate nut intake was also associated with better immediate and delayed memory in older adults with high risk of non-alcoholic fatty liver disease.
O’Neil et al. (2015) [69] United States	NHANES (2005–2010)	n = 14,386, 19+ years	Tree nut consumers comprised around 6% of the population. UI of tree nuts for consumers was 44.3 ± 1.6 g/day and per capita consumption was 3.3 ± 0.1 g/day.	Tree nut consumers had a significantly higher HEI-2005 and percentage of the population above the AI for fibre and potassium.
O’Neil et al. (2010) [68] United States	NHANES (1999–2004)	n = 13,292, 19+ years	Tree nut and tree nut butter consumption was low, with only 5.5% and 8.4% of participants 19–50 y and 51+ y consuming these.	Consumption of tree nuts was low but nutrient intake and diet quality significantly improved when tree nuts were consumed.
Jenab et al. (2006) [71] Europe	EPIC Cohort	n = 36,994	0.4% of men and 0.7% of women were pistachio consumers with average daily portion size intakes of 22 g/d and 23.1 g/d, respectively	There was a northern to southern European gradient of whole tree nut intake.

Key: EPIC, HEI, healthy eating index; HR, hazards ratio; NHANES, National Health and Nutrition Examination Survey; SD, standard deviation; UI, usual intake.

### 4.2. Recommended Intakes

Due to their health benefits, many guidelines recommend the daily consumption of nuts, except for the small percentage of individuals who have a nut allergy. In France the National Program for Nutrition and Health (PNNS) recommends the daily intake of a ‘small handful of unsalted nuts’ (a handful to reflect body size), highlighting their non-saturated lipid profile [74]. Pistachios are among the explicitly recommended nuts [74]. In the Netherlands, two tablespoons of nuts or seeds (15–25 g) daily are recommended as an alternative source of protein in the Wheel of Five [75].

Elsewhere, guidance throughout EU countries varies but where individual countries have formally considered nuts, the ensuing recommendations are usually around a handful of nuts and 25 g to 40 g/day are typically advised [75]. As described, the EAT-Lancet healthy reference diet also lists tree nuts advising a daily intake of 25 g (149 kcal), which includes pistachios, thus recognising these as a plant protein source [12].

In Australia and New Zealand, 30 g (or 1 ounce = 28.4 g) of nuts is advised as part of a cardioprotective diet [76,77]. In the United States the 2020–2025 Dietary Guidelines for Americans includes a protein subgroup including 5 oz equivalent of nuts, seeds, and soy per week at the 2000 calorie level (1 oz eq 28.4 g) although it should be recognised that this amount is not aligned with the 2003 heart health claim (1.5 oz (42.5 g) per day of nuts, such as pistachios) [70]. Canada’s Food guide recommends consuming plant-based sources of protein, including nuts, more regularly [78]. For further information related to nut consumption recommendations as part of dietary guidelines globally refer to Neale and Tapsell (2020) [79].

Overall, it is important to close the gap between intakes and recommendation to ensure a nutrient-rich plant-based diet. It can be observed that intake of tree nuts (including pistachios) is generally low, particularly amongst younger populations. Public health campaigns may be warranted to dispel myths, including that tree nuts are fattening, which could be resolved by more information [67]. Certainly, movements towards nutrient-rich, plant-based diets mean that populations should be considering dietary alignment with at least amounts specified in the EAT-Lancet healthy reference diet (25 g; 149 kcal).

## 5. Health Benefits of Pistachios

As described, according to European Commission regulations, pistachios are high in’ fibre, monounsaturates, copper, chromium, vitamin B6, thiamine, manganese, phosphorus and potassium and ‘a source’ of protein, vitamin E, K, folic acid, riboflavin, magnesium, iron, zinc and selenium. Pistachios also provide an array of vitamins and minerals alongside anthocyanins, carotenoids, flavonoids and phenolic acids [20,24].

They provide a spectrum of phenolic compounds, with at least 9 lipophilic and 11 hydrophilic bioactive being identified [80]. Most of the phenolics are present in the skin and the lipophilic constituents tend to be present in the nutmeat [80]. Research examining the phytochemical profile of American raw and roasted pistachios found that free-form contributions to the total phenolics were 82% and 84% for raw and roasted pistachios, respectively, and free-form contributions to the total flavonoids were 65% and 70% for raw and roasted pistachios, respectively [26]. For raw and roasted pistachios, gentisic acid and catechin were the predominant phenolics [26].

Polyphenols in pistachios are known to exert antioxidative and anti-inflammatory effects [20,81] and a range of publications have now documented their wider health effects [20,25,67,81,82].

### 5.1. Body Weight

Several studies have focused on inter-relationships between pistachio ingestion and markers of body weight and composition [83,84,85,86,87,88,89]. A randomised controlled study found that overweight/obese adults who ingested 42 g/day of pistachios alongside a weight-loss intervention had significantly increased intakes of fibre, ingested fewer sweets, and exhibited similar levels of weight loss to the control [83].

An extensive 24-week trial recruiting Asian Indians showed that daily consumption of unsalted pistachios (20% energy) significantly improved waist circumference and markers of metabolic syndrome (total-cholesterol, low-density lipoprotein, free fatty acids and adiponectin levels) [84]. Research with Chinese adults with metabolic syndromes similarly found that ingesting 42 g or 70 g pistachios daily for 12 weeks did not contribute to weight gain nor increased waist-to-hip ratio [85].

Research focusing on snacking, comparing the ingestion of 53 g of salted pistachios to 56 g of salted pretzels over 12 weeks by obese adults found that BMI declined significantly from 30.1 to 28.8 in the pistachio group compared with a 0.6 decline in the pretzel group [86]. Other work has shown that amongst healthy-weight women, 44 g (259 kcal) of pistachios daily can improve nutrient intake and induce satiety without impacting on body weight or composition [87,89]. Research by Bellisle *et al.* [89] showed that calories provided by pistachio snacks induced satiety and induced energy compensation in healthy women. As an afternoon snack, 56 g of pistachios (versus 56 g energy/protein matched savoury biscuits) did not influence body weight but did improve the micronutrient profile (thiamine, vitamin B6, potassium and copper) of French women, when consumed over four weeks [88].

Taken together, pistachios may help regulate body weight because of their satiety and satiation effects along with their reduced net metabolizable energy content [82]. Their fatty acid profile when consumed in moderation does not appear to impact on body weight and should be viewed as a heart-healthy fatty acid profile rather than one posing risk to gains in body weight.

### 5.2. Diabetes and Prediabetes

Pistachio constituents possess antioxidant and anti-inflammatory functions which may exert regulatory effects, including glucose- and insulin-lowering effects [20,21,90,91]. Kendall et al. (2011) [92] showed that pistachios (28, 56, and 84 g) added to white bread contributed to a dose-dependent reduction in glycaemic responses. The same team later showed that pistachio ingestion (versus white bread) reduced postprandial glycaemia and increased glucagon-like peptide levels [93]. A publication reanalysing data looking at nuts as a replacement for carbohydrates in diabetes diets further reconfirmed their role in improving glycaemic control in type 2 diabetes [94,95].

The Carlos Gestational Diabetes Mellitus (GDM) prevention study showed that early nutritional intervention with a Mediterranean diet that included pistachios reduced the risk of GDM and improved several maternal and neonatal outcomes, including rates of insulin-treated GDM, prematurity, excess gestational weight gain and rates of small and large-for gestational age infants [96,97,98]. Other research [99] conducted by Feng et al. (2019) investigated the acute effects of two isocaloric test meals—42 g of pistachios and 100 g of whole-wheat bread in Chinese women with GDM or gestational-impaired glucose tolerance. Pistachio intake resulted in significantly lower postprandial glucose, insulin and gastric inhibitory polypeptide and higher glucagon-like peptide-1 levels compared with whole-wheat bread, indicating that these would be a healthy snack choice during pregnancy [99].

Other factors such as telomere erosion have been linked to type 2 diabetes pathogenesis and severity [100]. A randomised crossover clinical trial allocated 49 prediabetic adults to consume a diet providing 57 g of pistachios daily or a calorie-matched control diet over four months with a two-week washout period [100]. The pistachio-supplemented diet significantly reduced DNA oxidative damage and upregulated telomerase reverse transcriptase (TERT) expression, which was inversely correlated to fasting plasma glucose levels [100]. Other work shows that similar intakes of pistachios (57 g/day) favourably alters microRNA expression linked to insulin sensitivity [101].

### 5.3. Heart Health

The U.S. FDA authorised the health claims that: “scientific evidence suggests but does not prove that eating 1.5 oz (42.5 g) per day of most nuts, such as pistachios, as part of a diet low in saturated fat and cholesterol may reduce the risk of heart disease” in 2003 [70].

Nuts are well recognised for their role in reinforcing heart health with a recent meta-analysis showing that nut consumption had a beneficial effect on reducing the incidence of (and mortality from) different cardiovascular disease outcomes [102]. A meta-analysis of 11 randomised controlled trials demonstrated that pistachio consumption improved cardiometabolic risk factors, including fasting blood sugar, insulin levels, systolic blood pressure and blood lipid profile [28].

A randomised trial comprised of 30 adults (40 to 74 years) demonstrated that pistachio nut consumption (replacement of low-fat or fat-free carbohydrate snacks with pistachios equivalent to 20% of daily energy (range: 59 to 128 g) depending on calorie assignment) improved certain cardiovascular risk factors, including heart rate variability and systolic blood pressure, with the latter observed most prominently during sleep [103]. Other early work [104] showed that diets providing 15% of calories as pistachio nuts (2–3 ounces; 57–85 g per day) improved certain lipid profiles in individuals with moderate hypercholesterolemia.

Ros et al. (2021) concluded that regular nut consumption is an indispensable component of any healthy, plant-based diet and that a daily dose of at least 30 g/d of a mixture of nuts, is ideal for optimising health [105]. The Global Burden of Disease Study (2017) estimated 21 g per day as the optimal intake of nuts and seeds after evaluating the health ramifications of suboptimal diets [106].

Subsequently, nuts appear to be an important dietary component for reinforcing heart health. Increasingly, plant-based dietary patterns are being viewed as beneficial for dyslipidaemia management, the prevention of cardiovascular disease risk and being environmentally sustainable [107].

### 5.4. Cancer

Tree nut intake has been shown to be associated with a significantly reduced incidence of colon cancer recurrence and mortality [108,109], pancreatic cancer [110,111] and overall cancer mortality [112,113,114]. Focusing on pistachios, in vitro and in vivo trials suggest that pistachio consumption could have a beneficial impact on cancer development [115]. Yuan et al. (2022) extracted free and bound phytochemical compounds in raw and roasted pistachios, finding that these demonstrated potent antioxidant and antiproliferative activities [26]. The free-form extracts of roasted pistachios exhibited relatively high antiproliferative capacity towards liver HepG2 (a human liver cancer cell line), along with colon Caco-2 and breast cancer MDA-MB-231 cells in a dose-dependent manner [26]. Of the extracts tested, roasted free extracts exhibited higher anticancer activities although free extracts of roasted pistachios had exceptionally high activity against human breast cancer MDA-MB-231 cells [26]. Other research by Glei et al. (2017) demonstrated chemopreventive potential of pistachio nuts using in vitro colon adenoma cells [115]. This effect was mediated by growth inhibition, the induction of apoptosis and anti-genotoxic effects, along with the induction of detoxifying enzymes [115]. Work has further shown that fermented pistachio milk possesses anti-colon cancer properties, which could be attributed to its acetate content [116].

### 5.5. Other Potential Benefits

Pistachios and pistachio extracts appear to play further roles in cognitive function, inducing neurobehavioral and neurochemical modifications [117,118] and exerting anxiolytic (anti-anxiety) effects [119,120]. Pistachio consumption also been linked to restoration of gut microbiota composition, with improved Bifidobacterium, Lactobacillus, Turicibacter and Romboutsia (beneficial bacteria) profiles [121,122]. Interestingly, after four weeks of prediabetic adults ingesting 57 g/day of pistachios, the urine profile of gut-microbiota metabolites altered significantly [123]. Further research has found that the ingestion of mixed tree nuts (1.5 oz over 12 weeks) affects tryptophan and microbial metabolism in overweight and obese subjects [124]. Changes in gut microbiota is an emerging area of research needing more attention.

## 6. Sustainability

Historically, meat and other animal-derived proteins have been regarded as being a superior protein source [13]. However, plant-based diets are gaining momentum and are now being viewed as a valuable protein source requiring fewer natural resources compared with animal-derived diets [13]. It is well understood that an animal-derived food-based system uses more energy, land and water resources than projected from lacto-ovo-vegetarian diets [125]. Dietary patterns higher in plant-based foods (e.g., nuts, seeds, legumes, vegetables, fruits and whole grains) and lower in animal-based foods are increasingly being found to be healthier and with fewer environmental ramifications [126].

A global review and meta-analysis have shown how transitioning towards healthier diets (including nuts) could reduce total/green water footprints [127]. The pistachio tree flourishes in dry climates and is well adapted to grow and thrive in semi-desert and desert areas of temperate and sub-tropical regions and therefore does not rely heavily on water as a resource [20]. Pistachio trees reach peak production at around 15 years and with proper care and management can last for decades, with Californian orchards projected to last for 70 to 100 years [128]. A publication examining water use for cultivation found that these trees survived well under drought conditions [129]. Yield was not significantly affected when deficit-irrigation restrictions were moderately applied during the growing season [129].

## 7. Discussion

The present publication has described the concept of protein quality. It has discussed how protein needs are changing—fuelled by growing and ageing global populations coupled with environmental degradation linked to traditional food consumption practices [17,130]. Future protein supplies subsequently cannot be ‘more of the same’ protein in higher proportions, hence, the concept of protein diversification is gaining attention along with interest in ‘reverting back’ to plant-based diets [16,17,18,131].

There have, however, been concerns that plant-based protein sources provide less protein overall, have poorer digestibility than animal protein and may result in inadequate intakes of protein and levels of EAAs compared to requirements [132]. Relevantly, modelling data on 1678 French adults showed that the introduction of legumes, nuts and seeds (LNS) into diets transitioning towards plant protein secured both protein intake and lysine adequacy; a finding attributed to their higher protein-to-energy ratios compared with cereals [132]. Similarly, other authors report that LNS are a sufficient means of achieving protein adequacy in adult vegetarians and vegans and that concerns over amino acid shortfalls are exaggerated [66]. Where the confusion appears to sit is when diets are insufficient in energy/protein and/or not providing LNS and that these instances could then hamper protein/amino acid and lysine intakes [66]. This is clearly a point that appears to be subject to miscommunication and thus in need of ‘ironing-out’ in public health campaigns.

As questioned by Katz et al. (2019) [14] we do not consume protein independently to other food sources; we ingest mixed diets with an array of amino acid profiles (as seen in Table 2). Given this, it is the amino acid profile of the overarching diet that will ultimately determine protein adequacy [14]. In 2019 Katz and co-authors concluded that “international organisations, such as WHO and FAO, periodically make recommendations for assessment of protein quality, as defined by how efficiently the consumption of a protein source can contribute to intake of essential amino acids” and that impacts on human health and the environment now need to be superimposed on top of this [14]. As explained, in the future, more than the PDCAAS needs to be measured. This method has its limitations and is not reflective of total ileal digestibility and does not apply health or environmental indices.

Pistachios are a source of good-quality plant protein, dietary fibre, abundant in vitamins (including A, C, E, K and folate), minerals (especially potassium, magnesium, calcium and phosphorus) and phytochemicals including carotenoids, tocopherols and phenolic compounds [67]. Furthermore, they qualify as a complete protein providing all nine EAAs with a DIAAS score (for raw and roasted pistachios) higher than 75, regarded as a good protein quality source [56]. The United States Department of Agriculture considers a serving of pistachios (28.35 g) to contain approximately 49 nuts [31]. As shown in Table 6 given the growing shifts in international recommendations for the global population to transition towards plant-based/whole-food diets, several points for consideration are needed.

Taken together, pistachios represent a nutrient-dense food that could be integrated within diets or eaten as a snack to meet future protein demands [25]. Alongside the protein that is provided from pistachios (a one-ounce serving; 28.35 g provides 5.7 g of quality, easily-digested protein) [31] their consumption has been linked to wide-ranging health benefits including cardio-metabolic health, glucose regulation and gut and cognitive function [20,21,22,27,28]. Eating pistachio nuts has not been associated with weight gain, as evidenced by several publications [25,86,87,89].

## 8. Conclusions

Taken collectively, protein demands are shifting. In fact, the transition towards plant-based protein consumption appears to be faster than the evolution of renewed approaches to determine protein quality. The present article has explained how the concept of ‘protein quality’ is now becoming obsolete. Consumers depend on accurate information to make informed food choices. Therefore, it is imperative that such methods are updated so that evidence-based messages integrating health and environmental factors can be communicated.

When focusing on the totality of ‘daily diets’, pistachio nuts have a key role to play in terms of providing ready-to-eat, good-quality, plant-based protein. They are a complete protein providing all nine EAAs and have a PDCAAS score of 73 and 81%, (raw and roasted pistachios, respectively). They also deliver an array of nutrients, phenolics, phytosterols and protective phytochemicals. Consuming pistachios can bring promising health benefits including greater potential of healthy glycaemic and appetite control, endothelial function, antioxidant, and inflammatory activity [22,25,27,82]. Their consumption also has potential to alleviate the pressure from other food protein production methods that are contributing to environmental strains.

## Figures and Tables

**Figure 1 nutrients-15-02158-f001:**
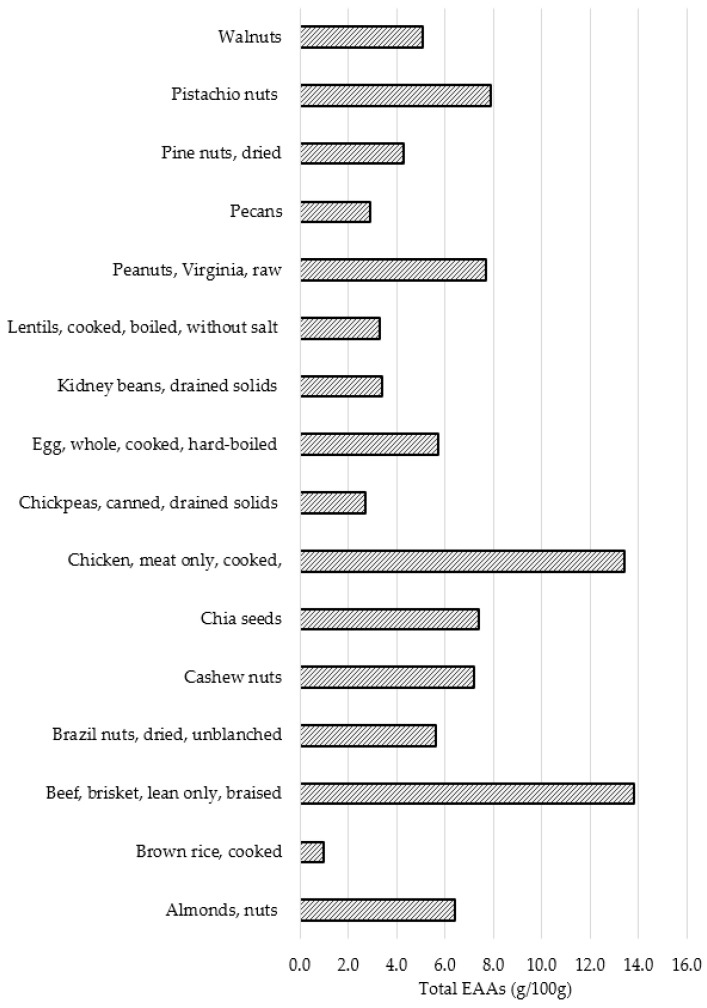
Total Essential Amino Acids (g/100 g) of Different Protein Foods.

**Table 2 nutrients-15-02158-t002:** International Protein Claims.

Country	Source of	High in/Good Source of/Rich in/Increased
Australia and New Zealand [51,52]	Good source—the food contains at least 10 g of protein per serving.	Increased (a) the food contains at least 25% more protein than in the same quantity of reference food; and (b) the reference food meets the general claim conditions followed by a nutrition content claim about protein.
EU [24]	≥12% energy from protein	≥20% energy from protein
UK [23]	≥12% energy from protein	≥20% energy from protein
US [53,54]	10–19% of the DRV of the corrected amount of protein ** per RACC. Equivalent to 5–9.5 g per RACC.	≥20% of the DV of protein per RACC *. Equivalent to ≥10 g protein per RACC *, **
Canada [42,55]	Protein rating of 20 or more ***	Protein rating of 20 or more and contains at least 25% more protein, totalling at least 7 g more than the food to which it is compared ***

Key: * RACC, reference amount customarily consumed; RACC for pistachio nuts = 30 g DRV (daily reference value); ** The “corrected amount of protein (gram) per serving” for foods is equal to the actual amount of protein (g) per serving multiplied by the PDCAAS; *** Protein rating (PR) is calculated as PR = estimated protein efficiency ratio for food × grams protein in a reasonable daily intake of the food. The “corrected amount of protein (g) per serving” for foods represented or purported for adults and children one or more years of age is equal to the actual amount of protein (g) per serving multiplied by the amino acid score corrected for protein for Canada using the PDCAAS as an alternate method of measuring protein quality [42,55].

**Table 3 nutrients-15-02158-t003:** PDCAAS Comparisons of Different Protein Foods.

Foods	PDCAAS (%)	Age for Standard Amino Acid Requirement	Reference Source
**Almonds, nuts, raw**	44–48%	2–5 years	House et al. (2019) [58]
**Almonds, Baru, roasted**	56.6%	2–5 years	Freita et al. (2012) [57]
**White rice, cooked**	56%	Healthy young men	Prolla et al. (2013) [60]
**Beef, dried beef, ground**	92.4%	2–5 years	Boye et al. (2012) [61]; Pires et al. (2006) [62]
**Brazil nuts, dried, raw**	63.3%	2–5 years	Freitas et al. (2012) [57]
**Cashew nuts**	90.3%	2–5 years	Freitas et al. (2012) [57]
**Chicken, fresh breast meat, dried**	95.2%	NCS	Negrão et al. (2005) [63]
**Chickpeas, canned, drained solids**	52%	NCS	Nosworthy et al. (2017) [64]
**Egg,** **lyophilised powder**	90.1%	2–5 years	Boye et al. (2012) [61] Pires et al. (2006) [62]
**Red** **kidney beans**	55%	--	Nosworthy et al. (2017) [64]
**Lentils,** **whole green**	63%	--	Nosworthy et al. (2017) [64]
**Lentils,** **split red**	54%	--	Nosworthy et al. (2017) [64]
**Lentils,** **split yellow**	64%	--	Nosworthy et al. (2017) [64]
**Peanuts,** **roasted**	69%	2–5 years	Freitas et al. (2012) [57]
**Pecans**	59%	--	Tanwar et al. (2022) [65] Calculated value *
**Pine** **nuts**	73%	--	Calculated value *
**Pistachio** **nuts, raw**	73%	2–5 years	Bailey et al. (2020) [56]
**Pistachio** **nuts, roasted**	81%	2–5 years	Bailey et al. (2020) [56]
**Walnuts**	39% 46%	6 months–3 years (Child) 3–10 years (Older child, adolescent, adult)	Lackey et al. (2021) [38]

Key: -- Not clearly specified. * Calculated value, from USDA amino acid content (USDA database [31]), using the percentage of the limiting amino acid (lysine) from the amino acid profile requirement for children 3 to 5 years of age (FAO [46,47]) and multiplying by true digestibility ([65]). Pistachio nuts highlighted in grey.

**Table 4 nutrients-15-02158-t004:** The Amino Acid Profiles of Different Protein Foods.

Amino Acids	Almonds, Nuts	Brown Rice, Cooked	Beef, Brisket, Lean Only, Braised	Brazil Nuts, Dried, Unblanched	Cashew Nuts	Chicken, Meat Only, Cooked, Grilled	Chickpeas, Canned, Drained Solids	Egg, Whole, Cooked, Hard-Boiled	Kidney Beans, Drained Solids	Lentils, Cooked, Boiled, without Salt	Peanuts, Virginia, Raw	Pecans	Pine Nuts, Dried	Pistachio Nuts	Walnuts
Alanine	1.0	0.2	1.8	0.6	0.8	1.8	0.3	0.7	0.4	0.4	1.0	0.4	0.7	**1.0**	0.7
Arginine	2.5	0.2	1.9	2.1	2.1	2.1	0.7	0.8	0.4	0.7	3.0	1.2	2.4	**2.2**	2.3
Aspartic acid	2.6	0.2	2.7	1.3	1.8	2.9	0.8	1.3	1.0	1.0	3.1	0.9	1.3	**2.0**	1.8
Cystine	0.2	0.0	0.4	0.3	0.4	0.3	0.1	0.3	0.1	0.1	0.3	0.2	0.3	**0.3**	0.2
Glutamic acid	6.2	0.5	4.5	3.2	4.5	4.5	1.2	1.6	1.3	1.4	5.3	1.8	2.9	**4.5**	2.8
Glycine	1.4	0.1	1.8	0.7	0.9	1.4	0.3	0.4	0.3	0.4	1.5	0.5	0.7	**1.1**	0.8
Histidine *	0.5	0.1	0.9	0.4	0.5	1.1	0.2	0.3	0.2	0.3	0.6	0.3	0.3	**0.5**	0.4
Isoleucine *^,b^	0.8	0.1	1.4	0.5	0.8	1.5	0.3	0.7	0.4	0.4	0.9	0.3	0.5	**1.0**	0.6
Leucine *^,b^	1.5	0.2	2.4	1.2	1.5	2.5	0.5	1.1	0.7	0.7	1.6	0.6	1.0	**1.7**	1.2
Lysine*	0.6	0.1	2.5	0.5	0.9	2.9	0.5	0.9	0.6	0.6	0.9	0.3	0.5	**1.2**	0.4
Methionine *	0.2	0.1	0.8	1.1	0.4	0.8	0.1	0.4	0.1	0.1	0.3	0.2	0.3	**0.4**	0.2
Phenylalanine *	1.1	0.1	1.2	0.6	1.0	1.2	0.4	0.7	0.5	0.4	1.3	0.4	0.5	**1.1**	0.7
Proline	1.0	0.1	1.4	0.7	0.8	1.0	0.3	0.5	0.5	0.4	1.1	0.4	0.7	**0.9**	0.7
Serine	0.9	0.1	1.2	0.7	1.1	1.2	0.4	0.9	0.5	0.4	1.2	0.5	0.8	**1.3**	0.9
Threonine *	0.6	0.1	1.2	0.4	0.7	1.4	0.3	0.6	0.3	0.3	0.9	0.3	0.4	**0.7**	0.6
Tryptophan *	0.2	0.0	2.0	0.1	0.3	0.4	0.1	0.2	0.1	0.1	0.2	0.1	0.1	**0.3**	0.2
Tyrosine	0.5	0.1	0.9	0.4	0.5	1.1	0.2	0.5	0.2	0.2	1.0	0.2	0.5	**0.5**	0.4
Valine *^,b^	0.9	0.2	1.4	0.8	1.1	1.6	0.3	0.8	0.5	0.4	1.0	0.4	0.7	**1.3**	0.8
BCAAs	3.2	0.5	5.2	2.5	3.4	5.6	1.1	2.6	1.6	1.5	3.5	1.3	2.2	**4.0**	2.6
EAAs	6.4	1.0	13.8	5.6	7.2	13.4	2.7	5.7	3.4	3.3	7.7	2.9	4.3	**8.2**	5.1
NEAAs	16.3	1.5	16.6	10.0	12.9	16.3	4.3	7.0	4.7	5.0	17.5	6.1	10.3	**13.8**	10.6
Data source/NDB Number	12,061	20,037	13,368	12,078	12,087	5747	16,358	1129	16,145	16,070	16,095	12,142	12,147	12,152	12,155

Key: ^b,^ BCAAs branched chain amino acids; * EAAs, essential amino acids; NEAAs, nonessential amino acids (highlighted in grey). Source: USDA Food Data Central (2022) [31].

**Table 6 nutrients-15-02158-t006:** Moving Forward – Some Considerations Towards Setting the Record Straight on Protein Quality.

It is advised that outlooks relating to ‘protein quality’ are refreshed and, as advised by Katz and colleagues (2019), updates are needed to account to health and environmental outcomes, which could be layered on top of PDCAAS or indeed other baseline scores reflective of protein quality [14].
Updated baseline measures of what constitutes ‘protein quality’ are needed as both the PDCAAS and DIAAS have their limitations, with the latter thought to produce erroneous results in the context of its application to plant-based diets [49,133].
It would make better sense to consider the ‘totality of diets’ and daily amino acids profiles rather than focusing in on individual, sole foods [14]. To facilitate the movement of science in this direction, future dietary surveys should collate intake data on separate amino acids alongside total daily protein intake.
The concept of protein diversification should begin to be uniformly expressed within national and international dietary guidelines [14,15,16,18].
Within future global dietary guidelines, nuts such as pistachios should be included and considered as a useful protein source. At present there is a lack of consensus. Some countries such as the U.S. [134] do specify nuts in their dietary guidelines but others do not.

## Data Availability

Not Applicable.

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
