# Peer review of "Believe It or ‘Nut’: Why It Is Time to Set the Record Straight on Nut Protein Quality: Pistachio (Pistacia vera L.) Focus"

_nutrients, 2023, doi:10.3390/nu15092158_

Round 1
Reviewer 1 Report
It is a very interesting narrative review. Except for a few typos errors, the language is so good. But I have some comments: 1) Could you please modify the title? 2) Is there any side effect of consuming pistachio? You mentioned only the positive health outcomes. If any, please incorporate them in the manuscript.
The quality of the English Language is so good except for few typos error.
Author Response
Thank you for your valuable feedback.
The manuscript has been proof-read again by all co-authors and a few typographical amendments made (highlighted in yellow).
The title has been modified so that the context is clearer.
Side effects have been highlighted/added - an allergy mention added and concerns that nuts are fattening.
Reviewer 2 Report
Plant-based diets are growing in popularity globally for an array of reasons including concerns for human and planetary health. This review provides an extensive and well-documented discussion of the quality of pistachio protein, a very interesting nut with a well-balanced composition. It takes into consideration the most up-to-date concepts of the nutritional quality of proteins. It considers as well the other phytochemicals in pistachio and their role in health. The text is well written and very easy to read.
Author Response
Thank you for the time taken to review the manuscript and the valuable feedback. We are glad that the manuscript was enjoyed.